# Adv-Attribute: Inconspicuous and Transferable Adversarial Attack on Face Recognition

**Shuai Jia**[1][†] **Bangjie Yin**[2][*] **Taiping Yao**[2] **Shouhong Ding**[2]
**Chunhua Shen**[3] **Xiaokang Yang**[1] **Chao Ma**[1][*]

[1] MoE Key Lab of Artificial Intelligence, AI Institute, Shanghai Jiao Tong University
[2] Youtu Lab, Tencent    [3] Zhejiang University
{jiashuai,xkyang,chaoma}@sjtu.edu.cn, jamesyin10@gmail.com
{taipingyao,ericshding}@tencent.com, chunhua@me.com

## Abstract

Deep learning models have shown their vulnerability when dealing with adversarial attacks. Existing attacks almost perform on low-level instances, such as pixels and super-pixels, and rarely exploit semantic clues. For face recognition attacks, existing methods typically generate the $\ell_p$-norm perturbations on pixels, however, resulting in low attack transferability and high vulnerability to denoising defense models. In this work, instead of performing perturbations on the low-level pixels, we propose to generate attacks through perturbing on the high-level semantics to improve attack transferability. Specifically, a unified flexible framework, Adversarial Attributes (Adv-Attribute), is designed to generate inconspicuous and transferable attacks on face recognition, which crafts the adversarial noise and adds it into different attributes based on the guidance of the difference in face recognition features from the target. Moreover, the importance-aware attribute selection and the multi-objective optimization strategy are introduced to further ensure the balance of stealthiness and attacking strength. Extensive experiments on the FFHQ and CelebA-HQ datasets show that the proposed Adv-Attribute method achieves the state-of-the-art attacking success rates while maintaining better visual effects against recent attack methods.

## 1 Introduction

Recent studies [1–6] have shown that deep learning based face recognition systems exhibit vulnerability to adversarial examples, which are constructed to fool models by adding perturbations to normal face images. A host of methods were developed to craft adversarial examples, and they realize to attack against white-box models [1], evaluate black-box model robustness [2], and create threats under physical scenarios to some extent [5–7].

However, most of these methods focus on generating pixel-level perturbations over the whole image. Considering that a human face is a typical combination of multiple significant parts, *e.g.*, the face form, the mustache, *etc*. Thus, the pixel-level attack may realize an imperceptible invade but lack crucial semantic clues [8], which lead to low transferability to other black-box models [4, 8]. Some other works attempt to generate wearable adversarial accessories [5–7], but such synthesized patches, *e.g.*, colorful glasses or distinct eyeshadows, are easily perceived by the observers. Recently, the work in [4] focuses on performing semantically meaningful perturbations in the visual attribute space. It directly interpolates the original faces and single attribute edited faces, which ignores the diverse

---

[*] Corresponding authors.
[†] Work done during an internship at Youtu Lab, Tencent.

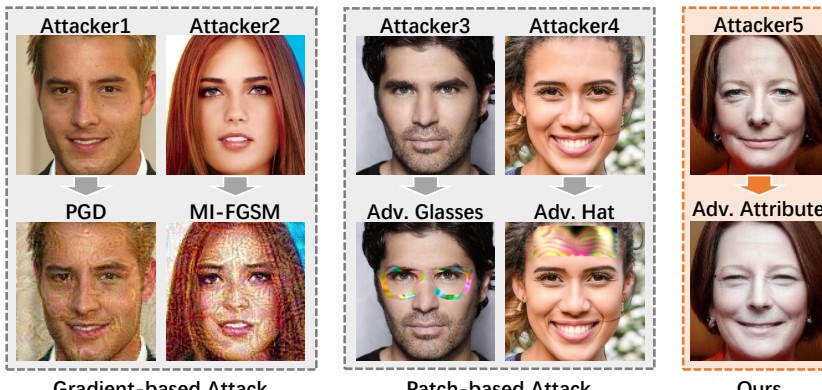

**Figure 1:** Illustration of the adversarial faces generated by Gradient-based attack, Patch-based Attack and our Adv-Attribute. The visual effects of the proposed Adv-Attribute are more natural and imperceptible.

contribution of different attributes when implementing the attack. Moreover, if the work in [4] utilizes several attributes together to generate adversarial examples, the adversarial faces are likely to have awful visual quality and are significantly different from the identity of clean attackers.

For stealthy and transferable attacks on face recognition, we argue that instead of performing perturbations on the low-level pixels, generating attacks through perturbing on the high-level semantics is more likely to improve attack transferability [4, 8]. Meanwhile, since stealthiness and attacking strength are contradictory to a certain degree, it is important to design a balanced strategy during adversarial attacks. On the one hand, stealthiness needs adversarial examples to stay with the original images closely. On the other hand, attacking strength requires adversarial examples to contain more perturbations.

To address these issues, we propose a novel adversarial attack method for face recognition, denoted as Adv-Attribute, which generates the adversarial noise and injects it into multiple attributes based on the guidance of the difference in face recognition features from the target. Concretely, we exploit the StyleGAN network as the face attribute editing tools and the disentangled attribute vectors from [9] as the perturbing ground-truth, which strongly maintains the visual effects of the generated faces and does not change the original identity. Moreover, since the importance of various facial attributes is different, we propose an importance-aware attribute selection strategy to adaptively select attribute vectors to perform attacks at each iteration. The dynamic selection of face attributes makes the generator learn the diverse attack strategies for various faces. Moreover, to further balance the stealthiness and transferability, the multi-objective optimization strategy is proposed by iteratively finding the Pareto-stationary solutions. In summary, our main contributions are as follows:

- We propose a novel unified attack method, termed Adv-Attribute, for face recognition systems by editing facial attributes, which crafts more imperceptible adversarial faces and highly improves the attack transferability as black-box attack.
- We further propose the importance-aware attribute selection and multi-objective optimization strategy to adaptively select attribute vectors to perform attacks and balance the attack stealthiness and transferability.
- Extensive experiments on the FFHQ and CelebA-HQ datasets demonstrate the effectiveness of our method against the state-of-the-art attack methods, and visualizations show that our adversarial faces are more inconspicuous.

## 2 Related Work

**Adversarial attack.** Many adversarial attack algorithms have indicated that deep learning models are broadly vulnerable to adversarial samples. For white-box attack, the gradient-based approaches [10–16] can be conducted by adding adversarial perturbations to the pixels of the original images, where all the perturbations are derived from the back-propagation gradients regarding to the adversarial constraints. For black-box attack, one interesting direction is to utilize a substitute/surrogate model to

perform transfer-based attacks. Zhou et al. [17] improve the adversarial transferability by attacking the middle layers of the surrogate models. Zhong et al. [2] explore to use a dropout strategy, introducing more randomness during adversarial noise generation, to improve the attack transferability. Recent works [2, 18, 19] claim that input diversity can further boost attack transferability. Another family of black-box attack methods simulate the targeted model by querying the model constantly. Authors of [20–22] propose a data-free approach to train a substitute model of the targeted black-box model, thus enabling attacks. Some works [23–28] propose to query the decision boundary or estimate gradients to perform black-box attacks. Nevertheless, those previous works do not consider the contradictory effects between imperceptible and adversarial constraints. To make adversarial examples look inconspicuously, their common approach is to apply the $\ell_p$ bound or manually tune the training weights. In the proposed Adv-Attribute, for the first time, we design a flexible multi-objective optimization paradigm to better balance the trade-off between stealthiness and attacking strength.

**Adversarial attack on face recognition.** One common attack for face recognition is gradient-based methods [10–14]. These attack methods add $\ell_p$-norm perturbations to the individual pixel, which decreases the attack transferability, and can be fragile to the denoising models. Besides, the above attack approaches in the digital world, performing attacks in physical scenarios has also been intensively studied. Komkov et al. [7] wear a printed adversarial hat and Sharif et al. [6] wear printed glasses as the physically-realizable forms respectively to attack the real-world face recognition models. However, these patch-based adversarial examples are easy to perceive and have weak transferability due to the limited editing region. Other works employ more stealthy attack approaches against face recognition models. Deb et al. [3] firstly use a GAN-based framework to synthesize face perturbations in the salient facial regions. Song et al. [1] also utilize GANs for crafting fake face images. Yin et al. [5] design a makeup generation framework to synthesize adversarial eye-shadow makeup to perform transferable attacks. Recently, Qiu et al. [4] explore a new attacking form, attribute-based adversarial attack. By using interpolation, they can encode adversarial clues into an individual face attribute. Note that the sample interpolation does not explore the usage of attributes and could yield a limited image quality. On the other side, these approaches still focus on a single attribute that is limited to the attacking performance. In contrast, our proposed Adv-Attribute creates adversarial noise by the combinations of multiple attributes compatibly and improves the stealthiness and attacking strength simultaneously.

## 3 Methodology

Given an original real face image $x$, adversarial attacks aim to search for a minimal perturbation $\epsilon$ to generate the adversarial face $\hat{x} = x + \epsilon$, where $\hat{x}$ is close to $x$ but misclassified by the target face recognition (FR) model $f(\hat{x})$. The optimization process can be expressed as the following objective function with constraints:

$$\min_{\epsilon} \lambda \parallel \epsilon \parallel_p + \mathcal{C}(f(\hat{x}), f(x)) , \text{ s.t. } \parallel \epsilon \parallel_p \leq c, \tag{1}$$

where $\mathcal{C}(\cdot, \cdot)$ denotes the adversarial criterion that uses the cosine distance loss between sources and targets for impersonation attack. Here $c$ is a constant to limit the magnitude of adversarial perturbations. Instead of applying the $\ell_p$-norm to $\epsilon$ proposed by those pixel-level based methods (*e.g.*, FGSM [10], PGD [12]), the proposed framework controls the perturbations $\epsilon$ by encoding the adversary into multiple facial attributes.

### 3.1 Adversarial Attributes Perturbation

We propose a novel framework, termed Adv-Attribute, by editing the adversary into facial images, which is based on the high-quality face editing method, StyleGAN [29]. A complete StyleGAN generator $G_s(\phi; g)$ consists of an encoder $\mathcal{E}$, a mapping network $\phi$ and a synthesis network $g$. By taking a disentangled latent vector $z \in \mathcal{Z}$ [9] for one particular attribute and a face image $x_o$ as the input, the StyleGAN generator synthesizes a new face image $\hat{x}_o$ with the edited attribute. For our attack method, we first obtain the attribute vectors $z_i$ from the disentanglement space $\mathcal{Z}$ similarly, where $i = \{1, ..., N\}$. Then we construct a vicinity appro vector $v_i = z_i + n_i$ by connecting the real attribute vectors $z_i$ with the adversarial noise perturbations $n_i$,

$$\hat{x}_o = G_a[m_o + \sum_{i}^{N}(z_i + n_i)], \tag{2}$$

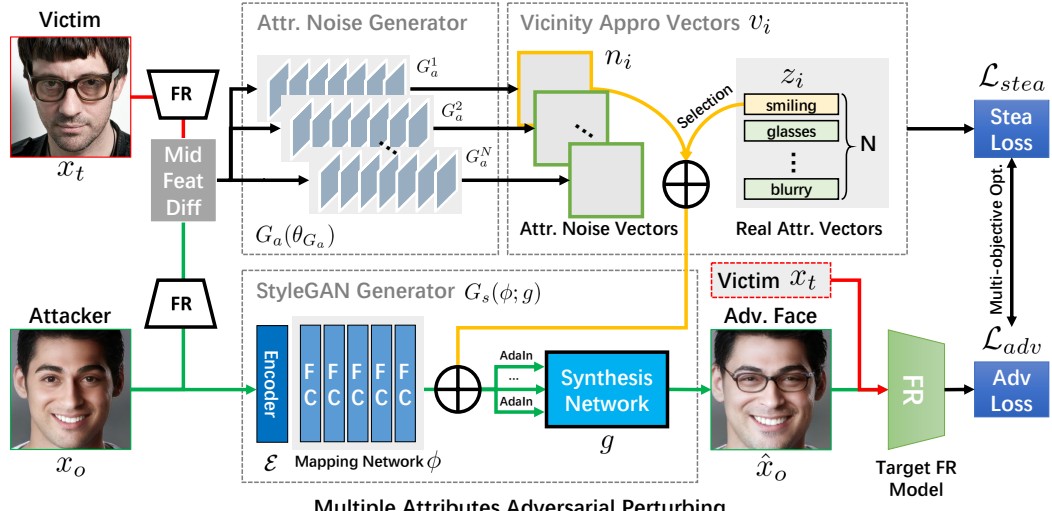

**Figure 2:** The overall framework of our proposed Adv-Attribute. We design a novel and unified pipeline to perturb the attributes of face images. Through this pipeline, we can generate adversarial faces with specially edited attributes, drastically improving the stealthiness of the attacks.

where $m_o$ is the original latent vector of $x_o$ generated by the encoder $\mathcal{E}$ and the mapping network $\phi$, *i.e.*, $m_o = \phi(\mathcal{E}(x_o))$, $N$ is the number of selected attributes, and $G_a$ is the adversarial noise generator (we will discuss it later). To ensure $v_i$ to better maintain the attribute information of the original $z_i$, we constrain the vicinity appro vector $v_i$ to the attribute vectors $z_i$. After that, we sent the vicinity appro vector $v_i$ into the StyleGAN generator $G_s(\phi; g)$ to generate the final adversarial face.

To search for an appropriate vicinity appro vector $v_i$ for each attribute latent vector $z_i$, we design an adversarial noise generator $G_a^i$ for each attribute. The input of $G_a$ is the difference $\kappa_{diff}$ between original attacking face $x_o$ and target victim face $x_t$,

$$\kappa_{diff} = \parallel f_m(x_o) - f_m(x_t) \parallel_p, \tag{3}$$

where the $f_m(\cdot)$ denotes the feature extraction in the middle layer of networks in the face recognition model. Adversarial noise vector $n_i$ will be computed as $n_i = G_a(\kappa_{diff})$. By taking the feature map difference as the input, $G_a$ can synthesize the $n_i$ guided by the prior of fine-grained semantic difference of the facial appearance.

The aim of impersonation attack is to fool the face recognition model to give higher similarity scores for two different identities. We use the cosine similarity loss as the impersonation attack loss,

$$\mathcal{L}_{adv} = 1 - cos[f(\hat{x}_o), f(x_t)], \tag{4}$$

where $f(\cdot)$ is the targeted FR model to extract the face embedding, $x_t$ is the target face and $\hat{x}_o$ is the adversarial face generated by the Style-GAN generator $G_s$. Moreover, we design the stealthy loss to add the constraint to the vicinity appro vector $v_i$ to enhance the stealthiness of the generated $\hat{x}_o$. The stealthy loss function can be defined as,

$$\mathcal{L}_{stea} = \frac{\sum_i^N [\alpha_1(1 - cos(z_i, v_i)) + \alpha_2 \parallel v_i \parallel_2]}{N}, \tag{5}$$

where $\alpha_1$ and $\alpha_2$ are used to adjust the weights between the cosine similarity and the $\ell_2$-norm of noises. The former item guarantees that the direction of $v_i$ is close to $z_i$, whereas the latter one ensures that the norm of $v_i$ is sufficiently small. We integrate these two losses into the overall training loss,

$$\mathcal{L}_{all} = \omega \cdot \mathcal{L}_{stea} + (1 - \omega) \cdot \mathcal{L}_{adv}, \tag{6}$$

where $\omega$ is the parameters to balance the weights of each loss function. Figure 2 illustrates the overall framework of the proposed attack method.

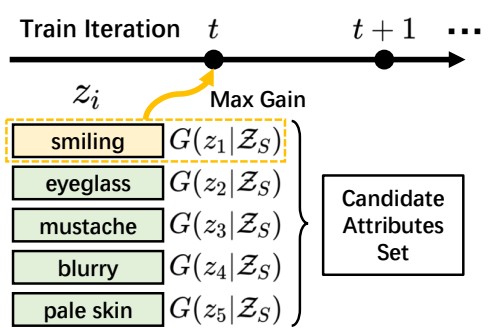
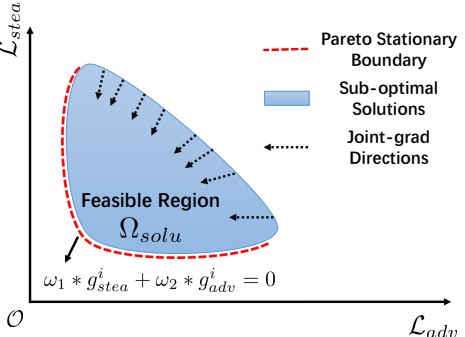

**Figure 3:** The illustration of importance-aware attribute selection strategy. During each step, it chooses the optimal attribute to make the encoding vector close to the one of target victim face.

**Figure 4:** An intuitive view of multi-objective optimization. During each step, it aims to find a dynamic optimizing balance between the impersonation attack loss and the stealthy loss.

## 3.2 Importance-Aware Attribute Selection

Editing the same attribute on diverse faces could play a different role on its identification. In other words, the importance of various facial attributes should be considered differently. Simply accumulating them together could give each attribute the equal steps of updating after $M$ iterations. We consider that this setting limits the adversarial potential for different attributes. Thus, we propose an importance-aware attribute selection strategy to adaptively select the training order based on its adversarial importance for different attribute noise generators. Inspired by [30, 31], the marginal gain of different attribute $z_i$ in each step can be expressed as:

$$G(z_i|\mathcal{Z}_S) = \mathcal{F}(z_i \cup \mathcal{Z}_S) - \mathcal{F}(\mathcal{Z}_S), \tag{7}$$

where $\mathcal{Z}_S$ denotes the attributes set without $z_i$ and $\mathcal{F}(\cdot)$ indicates the gain function to output the expected gain after editing $z_i$ into the face images. In our task, we adopt the $-\mathcal{L}_{adv}$ as the gain function and use $G(z_i|\mathcal{Z}_S)$ to measure the importance of various attributes. The corresponding attribute noise generator with the largest marginal gain will be selected to update in each step. By performing such a selection process, the updated steps for each attribute are diverse. If one attribute carries more adversary than others, its total number of selections will also be larger and thus the overall attack performance can be strengthened.

## 3.3 Multi-Objective Optimization

From the overall training loss $\mathcal{L}_{all}$, we consider that the stealthy loss $\mathcal{L}_{stea}$ and the adversarial loss $\mathcal{L}_{adv}$ are two conflicting objectives to some extent. Through the hand-crafted selection of their weights, it is difficult to obtain the optimal balance between two conflicting losses. According to the multi-objective optimization [32], the objective is to find a Pareto-stationary point to satisfy a KKT condition. The mathematical definition in our task can be written as:

$$\min_{\omega_1, \omega_2} \parallel \omega_1 \cdot g^i_{stea} + \omega_2 \cdot g^i_{adv} \parallel^2_2,$$
$$\text{s.t. } \omega_1 + \omega_2 = 1, \ \omega_1 \geq c_1, \omega_2 \geq c_2, \tag{8}$$

where $g^i = \nabla_{\theta_{G^i_a}} \mathcal{L}(\theta_{G^i_a})$ indicates the gradients with respect to the corresponding loss function, $\omega_1$ and $\omega_2$ in the solution space $\Omega_{solu}$ denote the trade-off parameters under our setting, $c_1$ and $c_2$ are the boundary constraints that determine the predefined bias. Ideally, we optimize $\omega_1, \omega_2$ as $\omega_1 \cdot g^i_{stea} + \omega_2 \cdot g^i_{adv} = 0$, where the gradient descent will not reduce the two conflicting loss functions. The solutions to $\omega_1$ and $\omega_2$ under this condition are the Pareto-stationary points. We hope that this multi-objective optimization can strengthen the transferability against the black-box FR models.

To solve this optimization problem, we denote $\hat{\omega}_1 = \omega_1 - c_1$, $\hat{\omega}_2 = \omega_2 - c_2$ and minimize the objective:

$$\min_{\hat{\omega}_1, \hat{\omega}_2} \parallel (\hat{\omega}_1 + c_1) \cdot g^i_{stea} + (\hat{\omega}_2 + c_2) \cdot g^i_{adv} \parallel^2_2,$$
$$\text{s.t. } \hat{\omega}_1 + \hat{\omega}_2 = 1 - (c_1 + c_2), \ \hat{\omega}_1 \geq 0, \hat{\omega}_2 \geq 0. \tag{9}$$

**Algorithm 1:** The proposed Adv-Attribute.

---

**Input:**    Source image $x_o \in \mathcal{X}_o$; Victim image $x_t \in \mathcal{X}_t$; Targeted FR model $f(\cdot)$;
                Pre-trained StyleGAN generator $G_s$; Adversarial noise generator $G_a^i, i \in \{1, ..., N\}$.
**Output:** Adversarial examples $x_{\text{adv}}$;

1  Initialize model parameters $\theta_{G_a} = \left\{\theta_{G_a^i}\right\}, i \in \{1, ..., N\}$ and trade-off weights $\omega_1, \omega_2$.
2  **for** $t = 1$ **to** $T$ **do**
3  $\quad$ Compute the feature difference $\kappa_{diff}$ through Eq. 3;
4  $\quad$ Take $\kappa_{diff}$ as the input of $G_a$, calculate noise vector $n_i$
   $\quad$ // Importance-aware attribute selection
5  $\quad$ Compute marginal gain $G(z_i|\mathcal{Z}_S)$ for each $z_i \in \mathcal{Z}$ via Eq. 7;
6  $\quad$ Select the optimal attribute $z_j$ with maximum gain;
   $\quad$ // Calculate the overall training loss
7  $\quad$ Calculate the adversarial loss $\mathcal{L}_{adv}^t$ via Eq. 4 and Eq. 2;
8  $\quad$ Calculate the stealthy loss $\mathcal{L}_{stea}^t$ via Eq. 5;
   $\quad$ // Multi-objective optimization
9  $\quad$ Compute $g_{stea}^{j(t)} = \nabla_{\theta_{G_a^j}^{t-1}} \mathcal{L}_{stea}(\theta_{G_a^j}^{t-1})$ and $g_{adv}^{j(t)} = \nabla_{\theta_{G_a^j}^{t-1}} \mathcal{L}_{adv}(\theta_{G_a^j}^{t-1})$ via Eq. 8;
10 $\quad$ Compute the balanced weights $\tilde{\omega}_1$ and $\tilde{\omega}_2$ by solving Eq. 9 and Eq. 11;
11 $\quad$ Update the weights $\omega_1 = \tilde{\omega}_1$ and $\omega_2 = \tilde{\omega}_2$;
   $\quad$ // Update the adversarial noise generator
12 $\quad$ Update $\theta_{G_a^j}^t \leftarrow \theta_{G_a^j}^{t-1} - \gamma[\omega_1 \cdot g_{stea}^{j(t)} + \omega_2 \cdot g_{adv}^{j(t)}]$;
13 **end**
14 **return** $\theta_{G_a}^* = \left\{\theta_{G_a^i}^T\right\}$

---

We first choose the equality constraints to form the relaxed problem. This solution can be given by the theorem in [32], applying the Lagrange multipliers and solving the Lagrangian problem. The formula can be expressed as:

$$\begin{pmatrix} \hat{\omega}_1^* \\ \hat{\omega}_2^* \\ \lambda \end{pmatrix} = (\mathbf{M}^T\mathbf{M})^{-1}\mathbf{M} \begin{bmatrix} -\mathbf{GG}^T\mathbf{c} \\ 1 - \mathbf{e}^T\mathbf{c} \\ \lambda \end{bmatrix}, \quad \mathbf{M} = \begin{bmatrix} \mathbf{GG}^T & \mathbf{e} \\ \mathbf{e}^T & 0 \end{bmatrix}, \tag{10}$$

where $\mathbf{G} = [g_{stea}^i, g_{adv}^i]$, $\mathbf{e} = [1, 1]$, $\mathbf{c} = [c_1, c_2]$ and $\lambda$ denotes the Lagrange multipliers. Secondly, we consider the non-negative constraints of $\hat{\omega}_1$ and $\hat{\omega}_2$. A non-negative least squares problem can be formulated as:

$$\min_{\tilde{\omega}_1, \tilde{\omega}_2} \| (\tilde{\omega}_1 - \hat{\omega}_1^*) + (\tilde{\omega}_2 - \hat{\omega}_2^*) \|_2^2,$$
$$\text{s.t.,} \ \tilde{\omega}_1 + \tilde{\omega}_2 = 1, \tilde{\omega}_1 \geq 0, \tilde{\omega}_2 \geq 0, \tag{11}$$

where the $\tilde{\omega}_1$ and $\tilde{\omega}_2$ are the optimized Pareto-stationary solutions for the multi-objective optimization. The whole training process of the proposed Adv-Attribute is presented in Algorithm 1.

## 4   Experiments

In this section, we first introduce the experimental setup. Then, we evaluate the performance of the proposed attack method on basic face recognition models and robust face recognition models with adversarial training. We also conduct ablation studies on the variants of our method. We finally evaluate the image quality of our method qualitatively and quantitatively.

### 4.1   Experimental Setup

**Datasets.** In our work, we choose two public facial datasets for evaluation: 1) FFHQ [29] is a high-quality human face dataset, which consists of more than 70,000 high-quality human face images with variations of age and ethnicity. 2) CelebA-HQ [33] is constructed as a higher-quality version of the CelebA dataset [34]. For each dataset, we randomly choose 100 faces as sources and another 10 faces as targets to construct 1000 source-target pairs for the impersonation attack.

| Method | Dataset | FFHQ | | | CelebA-HQ | | |
|---|---|---|---|---|---|---|---|
| | Targeted Model | IR152 | MobileFace | FaceNet | IR152 | MobileFace | FaceNet |
| Gradient-based | FGSM | 1.90 | 3.70 | 0.90 | 2.70 | 5.10 | 1.90 |
| | PGD | 19.30 | 21.80 | 2.60 | 26.00 | 29.90 | 3.50 |
| | MI-FGSM | 21.00 | 11.90 | 3.40 | 26.80 | 21.70 | 4.60 |
| | C&W | 16.30 | 16.20 | 2.00 | 27.30 | 28.20 | 3.30 |
| Patch-based | Adv-Hat | 13.60 | 3.10 | 4.80 | 2.50 | 8.40 | 4.70 |
| | Adv-Glasses | 2.50 | 4.60 | 8.20 | 4.50 | 5.60 | 9.10 |
| | Gen-AP | 12.00 | 19.90 | 8.20 | 19.50 | 24.40 | 15.80 |
| Stealthy-based | Adv-Face | 33.00 | 26.00 | 13.20 | 31.40 | 36.40 | 21.60 |
| | Adv-Makeup | 15.10 | 19.00 | 8.80 | 10.80 | 14.60 | 10.50 |
| | Semantic-Adv | 8.50 | 13.90 | 5.80 | 10.30 | 19.40 | 9.00 |
| Ours | w/o Selection | 44.10 | 49.10 | 30.70 | 43.70 | 46.40 | 28.60 |
| | w/o Multi-opt | 45.50 | 49.10 | 31.30 | 44.00 | 49.00 | 30.20 |
| | Adv-Attribute | **46.30** | **49.90** | **31.90** | **44.30** | **50.20** | **31.80** |

**Table 1:** *ASR* results of impersonation attack against basic models on the FFHQ and CelebA-HQ datasets. We choose three FR models (*i.e.*, IR152, MobileFace, and FaceNet) to evaluate the attack methods. For each column, we use two models to generate the adversarial faces to attack the other one as black-box attack.

**Evaluation metric.** For the evaluation metric, we adopt *attack success rate (ASR)* [3, 2] as:

$$ASR = \frac{\sum_i^N 1_\tau(cos[f(x_t^i), f(\hat{x}_o^i)] > \tau)}{N} \times 100\%, \tag{12}$$

where $1_\tau$ denotes the indicator function, $x_t$ and $\hat{x}_o$ are the target face and the generated face respectively, $\tau$ is the threshold and $N$ is the number of images. *ASR* aims to compute the proportion that the similarity score of source-target pairs is larger than $\tau$ in all source-target pairs.

**Implementation details.** The architecture of our face generator is mainly based on StyleGAN [35], which generates a high-quality semantic edited facial image in a high speed. We select five attributes (*i.e.,* smiling, eyeglass, mustache, blurry and pale skin) as our editing spaces. Note that the face attribute editing based on the original StyleGAN does not change the identity. During the attack, since we remain the StyleGAN model [35] fixed and assign no training data to the StyleGAN model, the quality of edited face images depends on the synthetic ability of StyleGAN. As for the noise generators $G_a$, we choose the layer before the FC/Pooling of various FR models as the middle feature. For instance, when attacking IR152, we extract the output of conv5_3 before the average pooling layer with the sizes of 512×7×7 to compute $\kappa_{diff}$. Since various FR models have different sizes of middle features, we adjust the shape of the parameters of FC layers as the last layer of $G_a$ to ensure the outputs match the real attribute size $z_i$ (*i.e.* 512×1). The attribute noise generators are trained using Adam optimizer [36] with an initial learning rate of 0.0001. The overall framework is implemented in PyTorch on one NVIDIA Tesla P40 GPU.

**Compared methods.** We compare our method with three types of attacks, gradient-based methods, patch-based methods and stealthy-based methods. Concretely, we select FGSM [10], PGD [12], MI-FGSM [13], and C&W [11] as the representative of gradient-based methods. For patch-based methods, we perform Adv-Hat [7], Adv-Glasses [6] and Gen-AP [8]. For stealthy-based methods, we choose Adv-Makeup [5], Adv-Face [3] and Semantic-Adv [4]. Different from the traditional $\ell_p$-norm where we can strictly guarantee that the perturbations will not exceed the bound, there is no strict guarantee that the perturbation will not exceed the attribute subspace in our method. For fair comparisons, we calculate the average variation of pixel values between original images and adversarial examples by our method. And we set the maximal perturbations as $\epsilon = 0.3$ for gradient-based methods. For patch-based methods, we do not constrain the magnitude by following its original setting.

## 4.2 Quantitative Results

**Attack on basic models.** For victim models, we first choose three representative FR models, IR152 [37], MobileFace [38] and FaceNet [39]. We train the Adv-Attribute framework with two FR models to generate the adversarial faces and perform them on the other FR model as black-box

| Dataset | FFHQ | | | | | | CelebA-HQ | | | | | |
|---|---|---|---|---|---|---|---|---|---|---|---|---|
| Training Model | -IR152 | | -MobileFace | | -FaceNet | | -IR152 | | -MobileFace | | -FaceNet | |
| Targeted Model | AT | TR | AT | TR | AT | TR | AT | TR | AT | TR | AT | TR |
| FGSM | 20.30 | 6.70 | 20.30 | 6.60 | 20.10 | 6.70 | 25.60 | 8.40 | 25.50 | 8.30 | 25.50 | 8.40 |
| PGD | 30.40 | 17.90 | 30.80 | 17.00 | 34.50 | 22.60 | 39.40 | 17.90 | 37.30 | 15.50 | 42.30 | 21.40 |
| MI-FGSM | 33.30 | 20.00 | 34.80 | 18.10 | 37.20 | 24.00 | 38.80 | 19.50 | 40.60 | 15.10 | 44.20 | 22.90 |
| C&W | 19.40 | 4.30 | 19.20 | 3.10 | 23.50 | 6.40 | 23.90 | 9.00 | 23.40 | 7.90 | 28.40 | 11.20 |
| Adv-Hat | 17.30 | 10.90 | 18.10 | 12.50 | 18.90 | 15.30 | 14.30 | 13.90 | 16.40 | 13.90 | 30.30 | 11.40 |
| Adv-Glasses | 24.30 | 16.00 | 21.00 | 12.00 | 15.50 | 10.70 | 24.70 | 12.20 | 30.00 | 17.10 | 24.70 | 12.80 |
| Gen-AP | 36.50 | 13.70 | 34.00 | 10.60 | 27.70 | 9.80 | 46.00 | 15.80 | 45.30 | 15.70 | 44.90 | 15.30 |
| Adv-Face | 37.00 | 21.80 | 31.40 | 17.20 | 34.00 | 14.80 | 54.00 | 19.60 | 53.60 | 13.60 | 47.00 | 8.00 |
| Adv-Makeup | 30.50 | 10.50 | 27.60 | 14.80 | 28.40 | 8.50 | 43.00 | 12.60 | 41.30 | 12.10 | 39.60 | 12.60 |
| Semantic-Adv | 12.30 | 4.00 | 12.30 | 3.80 | 12.40 | 3.80 | 15.30 | 3.80 | 15.40 | 4.00 | 15.50 | 4.00 |
| Ours | **60.80** | **33.50** | **56.60** | **34.00** | **61.60** | **34.10** | **60.40** | **30.10** | **53.30** | **26.30** | **63.00** | **30.10** |

**Table 2:** *ASR* results of impersonation attack against robust models on the FFHQ and CelebA-HQ datasets. During training, we still employ two of the three basic FR models to generate the adversarial faces. After that, we choose two FR robust models (*i.e.*, PDD-AT (AT), TRADES (TR)) to evaluate the attack methods. Note that '-' indicates the FR model that is not utilized in the training process.

| Dataset | FFHQ | CelebA-HQ |
|---|---|---|
| PGD | 113.57 / 75.52 | 94.35 / 77.95 |
| MI-FGSM | 118.00 / 95.47 | 97.81 / 95.67 |
| C&W | 123.29 / 99.90 | 112.57 / 96.62 |
| Adv-Face | 157.19 / 97.13 | 116.13 / 102.72 |
| Semantic-Adv | 138.16 / 95.98 | 127.29 / 97.10 |
| Ours | **74.86 / 69.55** | **68.52 / 70.38** |

**Table 3:** FID (↓) / MSE (↓) scores of different attack methods against black-box FR models on the FFHQ and CelebA-HQ datasets.

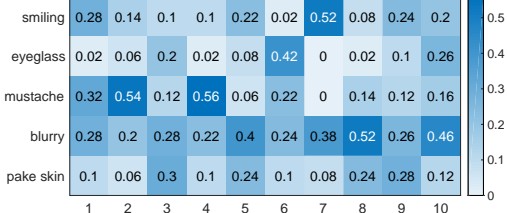

**Figure 5:** The frequency of each attribute for different source faces when attacking the same target face from the CelebA-HQ dataset.

attack. The value of $\tau$ in *ASR* is set as 0.01 FAR (False Acceptance Rate) for each victim FR model, *i.e.*, IR152 (0.167), MobileFace (0.302) and FaceNet (0.409). Table 1 reports the *ASR* results of impersonation attacks against basic models on the FFHQ and CelebA-HQ datasets. The *ASRs* of our method are significantly over 20% higher than the gradient-based methods and the patch-based methods. For the best stealthy-based method, Adv-Face, the *ASRs* of ours are still 23.9% and 13.8% higher for MobileFace on FFHQ and CelebA-HQ, respectively. Since the architecture of FaceNet is obviously different from the other two models, other methods have weak transferability on FaceNet. However, our method still can generate a robust adversary, leading to strong transferability.

**Attack on robust models.** In order to further illustrate the robustness of our attack method, we also choose two more robust FR models with adversarial training, PGD-AT [12] and TRADES [40] to evaluate the attack methods, as shown in Table 2. We follow their original thresholds $\tau$ (*i.e.*, PGD-AT (0.233) and TRADES (0.637) ) in *ASR* of these robust models. Most existing robust models are adopted by the gradient-based attack during adversarial training. Thus, these models have less impact on our attack method as we inject the adversary into the edited latent vector of attributes. Concretely, our method gets the *ASRs* of 60.80% and 33.50% for PGD-AT and TRADES on FFHQ and the *ASRs* of 60.40% and 30.10% for PGD-AT and TRADES on CelebA-HQ by training with the combination of MobileFace and FaceNet, which is considerably greater than other attacks. It indicates that our method also has strong attack transferability on the robust FR models with adversarial training.

**Ablation study.** To illustrate the effectiveness of each module in our attack method, we perform two variants of our method against basic models, *i.e.* ours w/o importance-aware attribute selection and ours w/o multi-objective optimization, as shown in Table 1. We observe that different types of attributes have diverse attack effects on face images and models. During each step, importance-aware attribute selection is applied to choose the optimal attribute that causes a maximal drop of adversarial loss. Figure 5 illustrates the frequency of five facial attributes on different source faces when attacking

| Target | Attacker | PGD | MI-FGSM | Adv-Glasses | Adv-Hat | Semantic-Adv | Ours |
|--------|----------|-----|---------|-------------|---------|--------------|------|

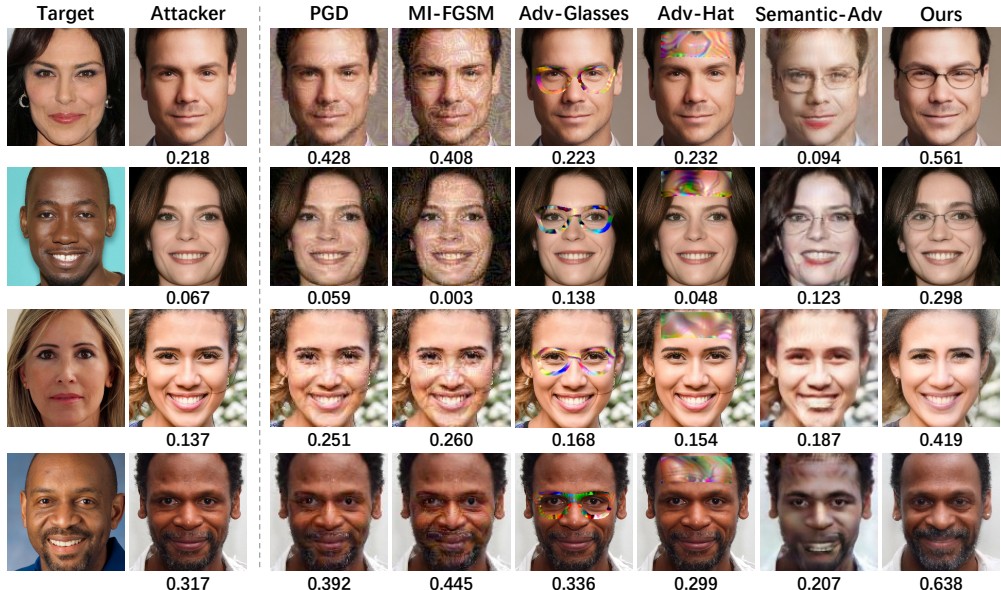

| | 0.218 | 0.428 | 0.408 | 0.223 | 0.232 | 0.094 | 0.561 |
| | 0.067 | 0.059 | 0.003 | 0.138 | 0.048 | 0.123 | 0.298 |
| | 0.137 | 0.251 | 0.260 | 0.168 | 0.154 | 0.187 | 0.419 |
| | 0.317 | 0.392 | 0.445 | 0.336 | 0.299 | 0.207 | 0.638 |

**Figure 6:** Visualizations of the adversarial faces generated by different attack methods. The number below the face images denotes the similarity score computed by MobileFace as black-box attack.

the same target face. It demonstrates that our method can adaptively seek a more optimal direction based on different face images and increase the attack ability slightly. On the other hand, there exists a trade-off between the image quality and the attack ability. A fixed parameter $\omega$ in Eq. 6 could deteriorate the image quality of some adversarial examples. Multi-objective optimization is used to find a dynamic optimizing balance between two conflicting losses, $\mathcal{L}_{stea}$ and $\mathcal{L}_{adv}$, which maintains a favorable balance between the attack success rate and image quality.

### 4.3 Image Quality Assessment

Figure 6 demonstrates the adversarial faces generated by various methods. Compared with gradient-based methods, the adversarial examples generated by our method have no obvious noise pattern as we inject the adversary into the latent vector. Compared with the patch-based method, our adversarial faces that only change the attributes are more natural and inconspicuous. When editing multiple attributes simultaneously, Semantic-Adv [4] directly interpolates the original faces and edited faces, whereas our method can adaptively choose the optimal attribute in each step and better complement the relationship of each attribute. In summary, our attack method generates more transferable and imperceptible adversarial faces. Furthermore, we use Frechet Inception Distance (FID) [41] and Mean Square Error (MSE) as metrics to evaluate the quality of the generated faces. For fair comparisons, we only choose the attack methods that modify the whole face. As shown in Table 3, the MSE scores indicate that the variation of our method on pixel-level is close to or less than other methods, but achieves better attack transferability. From the FID scores, they reveal that the quality of images generated by our method is significantly better than other methods quantitatively.

## 5 Conclusion

In this paper, we have proposed an inconspicuous and transferable adversarial attack method, Adv-Attribute, against face recognition. Different from the prior attack methods by adding the perturbations on the low-level pixels, our method injects the adversary into the edited latent vectors in several attributes. During the optimization, we propose an importance-aware attribute selection strategy to update the attribute noise with the large degradation of adversarial loss in each step. Meanwhile, we propose a multi-objective optimization to balance its stealthiness and attacking strength. Extensive experiments on the FFHQ and CelebA-HQ datasets indicate that the proposed attack method Adv-Attribute has strong attack transferability across different face recognition models and yields natural and inconspicuous adversarial faces. In the future, we will further study such physical affections and design a discriminative and robust face recognition model.

## Negative Societal Impacts

The proposed method may be used maliciously to hazard the security of existing face recognition models in real life. Hopefully, our work will draw attention to the safety issues of face recognition by improving their robustness.

## Acknowledgment

This work was supported by NSFC (61906119, U19B2035, 62171281), Shanghai Municipal Science and Technology Major Project (2021SHZDZX0102), and CCF-Tencent Open Research Fund. C. Shen's participation was in part supported by a major grant from Zhejiang Provincial Government.

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
