# Adv-Attribute: Inconspicuous and Transferable Adversarial Attack on Face Recognition
## —Supplementary Material—

In this supplementary material, we provide more details and experimental results to complement the manuscript. Firstly, we report the frequency of different attributes when attacking diverse targets with our importance-aware attribute selection. Secondly, we plot the variations of the overall loss with or without our multi-objective optimization during optimization. Thirdly, we provide the complete proof of Pareto-stationary solution. Finally, we illustrate the comparison of edited faces by the original StyleGAN [1] and the proposed Adv-Attribute attack.

## A  Important-aware Attribute Selection

When editing the same source face, different attributes could play a different role on the identification of FR models. During training, the proposed important-aware attribute selection can choose the optimal attribute for the different pairs of target faces and source faces. Due to the limited page in the submission, we supply more frequency of different attributes when attacking different target faces from FFHQ [2] and CelebA-HQ [3], as illustrated in Figure A. We report the frequency of five attributes (*i.e.,* smiling, eyeglass, mustache, blurry and pale skin) between different source faces and different target faces. When attacking the same target face, diverse source faces choose different attributes in each step. In other words, the same source face also selects different attributes for different target faces.

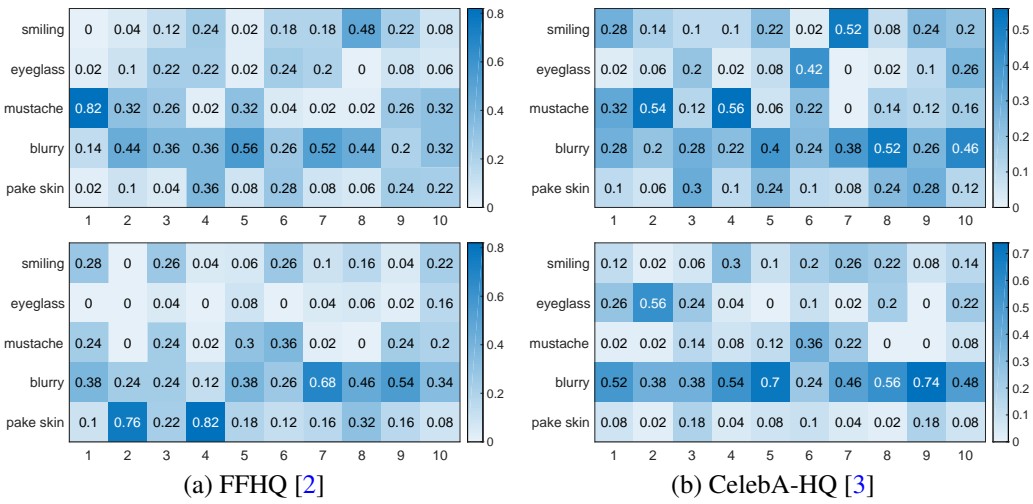

(a) FFHQ [2]  (b) CelebA-HQ [3]

Figure A: The frequency of each attribute for different source faces when attacking the same target face from the FFHQ [2] and CelebA-HQ [3] datasets.

## B  Multi-Objective Optimization

During the training process, Figure B plots the variation of the overall loss with or without the multi-objective optimization. As the training steps increase, $\omega_1$ and $\omega_2$ are dynamically adjusted to balance the weights of the impersonation attack loss $\mathcal{L}_{adv}$ and the stealthy loss $\mathcal{L}_{stea}$ with our strategy,

whereas we set $\omega_1 = 0.5$ and $\omega_2 = 0.5$ fixed when applying our attack without the multi-objective optimization. Although both the overall losses $\mathcal{L}_{all}$ are decreasing with the increase of epochs, our method combined with the multi-objective optimization achieves a more efficient training approach and yields a stronger attack.

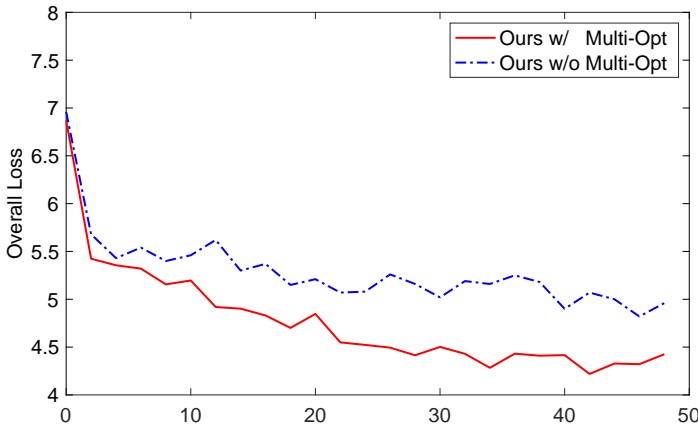

Figure B: Comparison of the overall training loss with or without the multi-objective optimization.

## C   Proof of Pareto-stationary Solution

**Lemma 1.** Suppose the overall training loss $\mathcal{L}_{all}$ consists of two conflicting objectives, the stealthy loss $\mathcal{L}_{stea}$ and the adversarial loss $\mathcal{L}_{adv}$. According to the multi-objective optimization [4], the aim is to optimize the trade-off parameters $\omega_1$ and $\omega_2$ to satisfy as:

$$\min_{\omega_1,\omega_2} \parallel \omega_1 \cdot g^i_{stea} + \omega_2 \cdot g^i_{adv} \parallel^2_2,$$
$$\text{s.t. } \omega_1 + \omega_2 = 1, \ \omega_1 \geq c_1, \omega_2 \geq c_2, \tag{1}$$

where $g^i$ indicates the gradients with respect to the corresponding loss function, $c_1$ and $c_2$ are the boundary constraints that control the predefined bias. The solutions to $\omega_1$ and $\omega_2$ that $\omega_1 * g^i_{stea} + \omega_2 * g^i_{adv} = 0$ are the Pareto-stationary points under a KKT condition:

$$\begin{pmatrix} \hat{\omega}_1^* \\ \hat{\omega}_2^* \\ \lambda \end{pmatrix} = (\mathbf{M}^T\mathbf{M})^{-1}\mathbf{M} \begin{bmatrix} -\mathbf{G}\mathbf{G}^T\mathbf{c} \\ 1 - \mathbf{e}^T\mathbf{c} \\ \lambda \end{bmatrix}, \ \mathbf{M} = \begin{bmatrix} \mathbf{G}\mathbf{G}^T & \mathbf{e} \\ \mathbf{e}^T & 0 \end{bmatrix}, \tag{2}$$

where $\mathbf{G} = [g^i_{stea}, g^i_{adv}]^T$, $\mathbf{e} = [1, 1]^T$, $\mathbf{c} = [c_1, c_2]^T$ and $\lambda$ denotes the Lagrange multipliers.

**Proof of Lemma 1.** To solve this optimization problem, we denote $\hat{\omega}_1 = \omega_1 - c_1$, $\hat{\omega}_2 = \omega_2 - c_2$ and minimize the objective:

$$\min_{\hat{\omega}_1,\hat{\omega}_2} \parallel (\hat{\omega}_1 + c_1) \cdot g^i_{stea} + (\hat{\omega}_2 + c_2) \cdot g^i_{adv} \parallel^2_2,$$
$$\text{s.t. } \hat{\omega}_1 + \hat{\omega}_2 = 1 - (c_1+c_2), \ \hat{\omega}_1 \geq 0, \hat{\omega}_2 \geq 0. \tag{3}$$

The equality constraints can be formulated as a relaxed problem, which can be written as:

$$\min_{\hat{\omega}} \frac{1}{2}\hat{\omega}^T\mathbf{G}\mathbf{G}^T\hat{\omega} + \mathbf{c}^T\mathbf{G}\mathbf{G}^T\hat{\omega} + \frac{1}{2}\mathbf{c}^T\mathbf{G}\mathbf{G}^T\mathbf{c}, \ \text{s.t. } \mathbf{e}^T\hat{\omega} = 1 - \mathbf{e}^T\mathbf{c}, \tag{4}$$

where $\hat{\omega} = [\hat{\omega}_1, \hat{\omega}_2]^T$, $\mathbf{G} = [g^i_{stea}, g^i_{adv}]^T$, $\mathbf{e} = [1, 1]^T$ and $\mathbf{c} = [c_1, c_2]^T$.

After that, we use the Lagrange multipliers and solve the Lagrangian as:

$$\mathcal{L}(\hat{\omega}, \lambda) = \frac{1}{2}\hat{\omega}^T\mathbf{G}\mathbf{G}^T\hat{\omega} + \mathbf{c}^T\mathbf{G}\mathbf{G}^T\hat{\omega} + \lambda\mathbf{e}^T\hat{\omega} - 1 + \mathbf{e}^T\mathbf{c}. \tag{5}$$

The partial derivatives with respect to $\hat{\omega}$ and $\lambda$ are computed to find the Pareto-stationary points,

$$\frac{\partial\mathcal{L}(\hat{\omega}, \lambda)}{\partial\hat{\omega}} = 0, \ \frac{\partial\mathcal{L}(\hat{\omega}, \lambda)}{\partial\lambda} = 0. \tag{6}$$

| Target face | Original face | Original StyleGAN | Our attack |
|:---:|:---:|:---:|:---:|

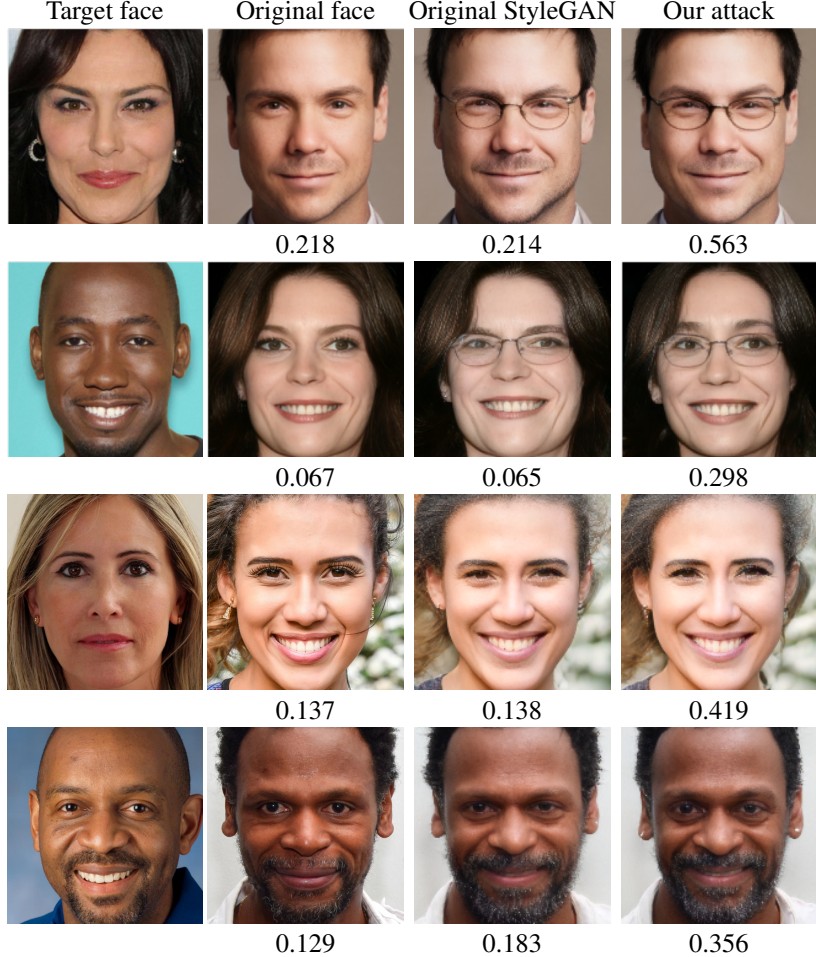

| | 0.218 | 0.214 | 0.563 |
|:---:|:---:|:---:|:---:|
| | 0.067 | 0.065 | 0.298 |
| | 0.137 | 0.138 | 0.419 |
| | 0.129 | 0.183 | 0.356 |

Figure C: Comparison of the original source faces, the edited faces by original StyleGAN [1] and the edited faces by our attack (from the left to right). The number below the face images denotes the similarity score computed by MobileFace [6] as black-box attack.

The optimization of this problem can be solved as:

$$\mathbf{M} \begin{pmatrix} \hat{\omega}_1^* \\ \hat{\omega}_2^* \\ \lambda \end{pmatrix} = \begin{bmatrix} -\mathbf{G}\mathbf{G}^T\mathbf{c} \\ 1 - \mathbf{e}^T\mathbf{c} \\ \lambda \end{bmatrix}. \tag{7}$$

According to the Moore-Penrose inverse [5], the final results are expressed as:

$$\begin{pmatrix} \hat{\omega}_1^* \\ \hat{\omega}_2^* \\ \lambda \end{pmatrix} = (\mathbf{M}^T\mathbf{M})^{-1}\mathbf{M} \begin{bmatrix} -\mathbf{G}\mathbf{G}^T\mathbf{c} \\ 1 - \mathbf{e}^T\mathbf{c} \\ \lambda \end{bmatrix}, \quad \mathbf{M} = \begin{bmatrix} \mathbf{G}\mathbf{G}^T & \mathbf{e} \\ \mathbf{e}^T & 0 \end{bmatrix}. \tag{8}$$

## D   Comparison with Original StyleGAN

To study the effect of identity when editing the faces by the original StyleGAN [1], we choose 100 face images from FFHQ and CelebA-HQ and randomly select from these five attributes with different magnitudes to edit the faces ten times. Meanwhile, we calculate the average recognition accuracy between original faces and edited faces by original StyleGAN. We find that the recognition accuracy is 100% for all three FR models (*i.e.,* IR152, MobileFace and FaceNet) on both datasets, which indicates that editing facial attributes with StyleGAN [1] does not change its original identity.

| Dataset | FFHQ | | | CelebA-HQ | | |
|---|---|---|---|---|---|---|
| FR Model | IR152 | MobileFace | FaceNet | IR152 | MobileFace | FaceNet |
| Original faces | 0.045 | 0.168 | 0.083 | 0.052 | 0.179 | 0.099 |
| Edited faces by original StyleGAN | 0.043 | 0.147 | 0.061 | 0.049 | 0.157 | 0.072 |
| Edited faces by our attack | 0.231 | 0.306 | 0.412 | 0.237 | 0.306 | 0.429 |

Table A: Cosine similarity between target faces and original faces, edited faces by original StyleGAN and edited faces by our attack.

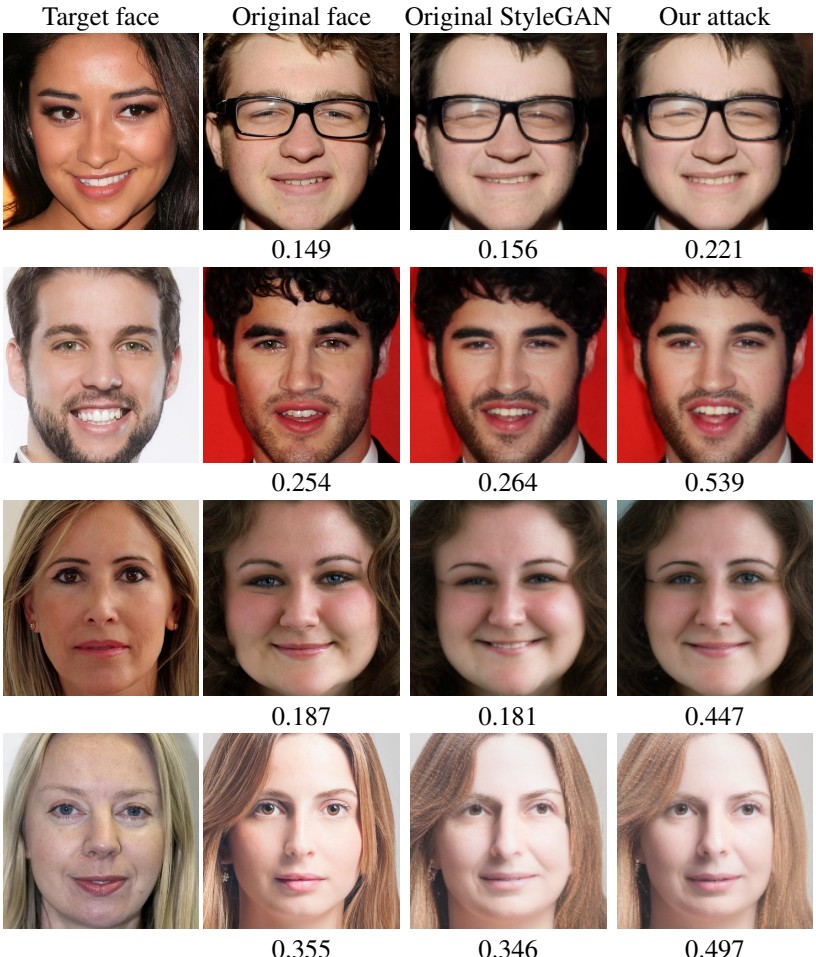

Figure D: Comparison of the original source faces, the edited faces by original StyleGAN [1] and the edited faces by our attack (from the left to right).

On the other hand, the aim of our attack is to make FR models recognize the adversarial faces as the targeted person. We compute the average cosine similarity between target faces and original faces, edited faces by original StyleGAN and edited faces by our attack on FFHQ [2] and CelebA-HQ [3], as reported in Table A. Note that the edited face images by original StyleGAN [1] are crafted without the proposed noise generators and we control the editing level by original StyleGAN [1] as close to our attack method. These results indicate that the original attribute editing hardly affects the similarity to the target face, whereas the adversarial faces by our attack successfully impersonate the targeted identity with higher cosine similarity. Additionally, Figure C illustrates a qualitative comparison between edited faces by original StyleGAN and edited faces by our Adv-Attribute attack. Not only that the adversarial faces by our method are close to the editing faces by original StyleGAN [1], but our attack also increases the similarity score with target faces to achieve impersonation attacks.

| Target face | Original face | Original StyleGAN | Our attack |
|---|---|---|---|

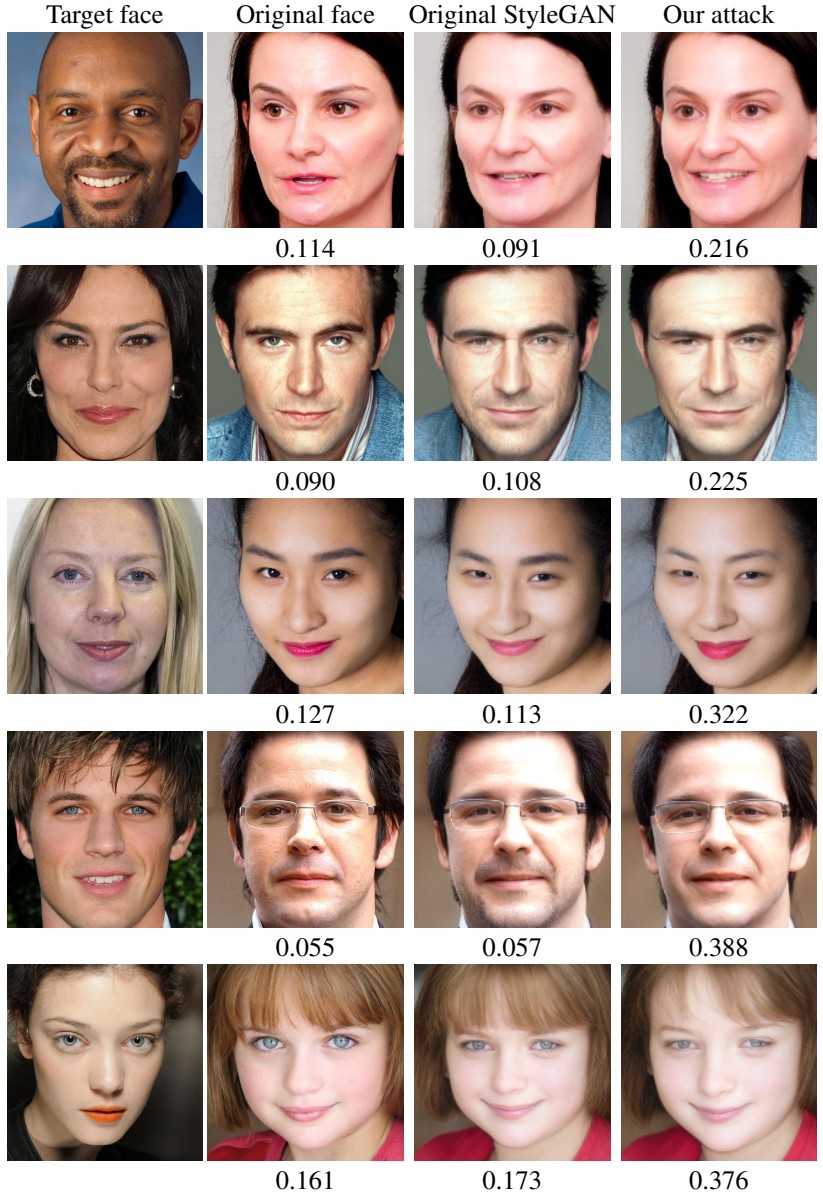

|  | 0.114 | 0.091 | 0.216 |
|  | 0.090 | 0.108 | 0.225 |
|  | 0.127 | 0.113 | 0.322 |
|  | 0.055 | 0.057 | 0.388 |
|  | 0.161 | 0.173 | 0.376 |

Figure E: Comparison of the original source faces, the edited faces by original StyleGAN [1] and the edited faces by our attack (from the left to right).

## E   More Qualitative Results

This section provides more qualitative results from FFHQ [2] and CelebA-HQ [3]. Figure D and Figure E compare the original source faces, the edited faces by original StyleGAN [1] and the edited faces by our attack. In general, the majority of adversarial edited faces achieve favorable visual quality and attacking performance, while only several examples (e.g., the second row in Figure E) are slightly semantically-inconsistent with the original image for human observers. Moreover, since we choose pale skin and blurry as two of five selected attributes in our editing spaces, these two attributes could smooth and lose partial textures on edited faces by both the original StyleGAN and our attack. The synthetic ability of StyleGAN decides the quality of edited face images, serving as a potential weakness of our attack.