# OpenReview forum: "Adv-Attribute: Inconspicuous and Transferable Adversarial Attack on Face Recognition"
_NeurIPS.cc/2022/Conference — NeurIPS 2022 Accept_

### Official Review · Reviewer_wjfx · 2022-07-09

**Rating:** 7
**Confidence:** 4
**Soundness:** 4 excellent
**Presentation:** 3 good
**Contribution:** 4 excellent

**Summary:**

This paper proposes an adversarial attack method, Adversarial Attributes, against face recognition. To improve attack transferability, this paper generates attacks on the high-level semantics by injecting the adversary into the edited latent vectors in several attributes. To balance stealthies and attacking strength, an importance-aware attribute selection and the multi-objective optimization strategy are introduced. Experiments carried out on FFHQ and CelebA-HQ datasets demonstrate the effectiveness of the proposed method.

**Questions:**

1. Which equation corresponds to the vicinity appro vector vi? How to define the vicinity appro vector vi statistically?

2. What is the distribution of values of balanced weights w1 and w2? Could you show more results for these two balanced weights?

3. Why most of textures of adversarial faces generated by the proposed attack methods are smoothed and lost?


**Limitations:**

Yes

**Strengths And Weaknesses:**

Strengths:

1. The idea of generating attacks through perturbing on the high-level semantics (facial attributes) is interesting and novel and facilitates attack transferability.

2. Proposed Adversarial Attributes Perturbation has some technical novelties with clear figure illustration. Importance-Aware Attribute Selection and Multi-Objective Optimization is reasonable with theoretical support.

3. Evaluation is reasonably thorough and both promising qualitative and quantitative results are claimed. Ablation study also demonstrates the effectiveness of each module in the proposed attack method.

Weaknesses：

From Visualizations in this paper, e.g. Fig. 6, it seems textures of adversarial faces generated by the proposed attack methods are smoothed and lost. This may be one of the drawbacks of the proposed method.

---

> ### Author Response · Authors · 2022-08-01
> **Response to Reviewer wjfx**
>
> Thanks for your valuable comments and we address your concerns as follows.
>
> **[Q1: Vicinity appro vector $v_i$.]** The definition of the vicinity appro vector $v_i$ corresponds to Eq.2 as $v_i = z_i + n_i $. We will add this equation in the revision.
>
> **[Q2: Distribution of $\omega_1$ and $\omega_2$.]** We illustrate the variation of balanced weights $\omega_1$ and $\omega_2$ when attacking IR152 on the CelebA-HQ dataset in the table below.
>
> |  Epoch   | 1  | 5 | 10| 15 | 20 | 25 | 30 | 35 | 40 | 45 | 50 |
> |  :----:  | :----:  | :----:  | :----:  | :----:  | :----:  | :----:  | :----:  | :----:  | :----:  | :----:  | :----:  |
> | $\omega_1$  | 0.50 | 0.56 | 0.72 | 0.66 | 0.35 | 0.84 | 0.77 | 0.61 | 0.47 | 0.43 | 0.75 |
> | $\omega_2$  | 0.50 | 0.44 | 0.28 | 0.34 | 0.65 | 0.16 | 0.23 | 0.39 | 0.53 | 0.57 | 0.25 |
> | $L_{all}$        | 6.87 | 5.36 | 5.16 | 4.92 | 4.70 | 4.52 | 4.41 | 4.42 | 4.42 | 4.32 | 4.43 |
>
> As the number of epochs increases, $\omega_1$ and $\omega_2$ are dynamically adjusted to balance the weights of the impersonation attack loss $L_{adv}$ and the stealthy loss $L_{stea}$, resulting in a drop of the overall training loss $L_{all}$. Additionally, Figure B in supplementary material plots the variation of the overall loss $\mathcal{L}_{all}$ with and without the multi-objective optimization (i.e., set $\omega_1=0.50$ and $\omega_2=0.50$ fixed during training), which indicates that our method combined with the multi-objective optimization better balances these two conflicting losses and yields a stronger attack.
>
>
> **[Q3: Textures of adversarial faces.]** During our attack, we keep the face generator (i.e., StyleGAN [19]) fixed to edit semantic attributes on face images. Thus, the quality of edited face images is limited by the synthetic ability of StyleGAN [19]. In supplementary material, Figure C compares original source images, edited faces by the original StyleGAN, and edited faces by our attack. The image quality of edited faces by our attack is close to the ones edited by the original StyleGAN. Moreover, since we choose pale skin and blurry as two of five selected attributes in our editing spaces, these two attributes could smooth and lose partial textures on edited faces by both the original StyleGAN and our attack. We will add more discussion in the revision.

---

### Official Review · Reviewer_yMkx · 2022-07-10

**Rating:** 7
**Confidence:** 4
**Soundness:** 3 good
**Presentation:** 3 good
**Contribution:** 3 good

**Summary:**

The authors have proposed to produce inconspicuous and transferable adversarial attacks on face recognition systems. Instead of perturbing the pixel intensity, the authors propose to semantically perturb the facial attributes such as smiling, eyeglass, mustache, blurry, and pale skin. Although adversarially manipulating facial semantics or attributes is not new, in this paper, the authors do bring to the table something new: (1) importance-aware attribute selection strategy that can select and update one particular attribute noise vector that leads to the largest degradation of adversarial loss in each step, and (2) multi-objective optimization that can balance the stealthiness and attacking strength. Experimentally, the proposed method is benchmarked against several types of adversarial attacks including gradient-based noise attacks, patch-based attacks, and stealthy-based attack methods. The evaluation is carried out on both basic face recognition models, as well as robust ones with adversarial training. The proposed method performs favorably compared to the baselines in terms of both the direct attack success rate as well as black-box transferability.


**Questions:**

I do not have questions.


**Limitations:**

Yes.

**Strengths And Weaknesses:**

Strengths:

(1) The paper is well written. The presentation and organization of various components is very clear. The experiments are thoroughly carried out with adequate baselines as a comparison.

(2) Although adversarial perturbation based on semantic manipulation is not new, the authors have managed to incorporate two new components (importance-aware attribute selection and multi-objective optimization to balance the two conflicting losses) to improve upon prior semantic-based adversarial face perturbations.

(3) The experimental results are pretty strong, in favor of the proposed method, in terms of the performance on both the basic FR, robust FR, as well as on black-box transferability tasks.

Weaknesses:

This is overall a solid paper. At this review stage, I do not seem to spot apparent weaknesses in this submission.


Minor point: on line 206, there is a typo: steady-based should be stealthy-based.

---

> ### Author Response · Authors · 2022-08-01
> **Response to Reviewer yMkx**
>
> Thanks for your careful review. We will fix this typo and thoroughly proofread the paper again.

---

### Official Review · Reviewer_aXGo · 2022-07-11

**Rating:** 5
**Confidence:** 4
**Soundness:** 3 good
**Presentation:** 3 good
**Contribution:** 2 fair

**Summary:**

This paper presents a new attack based on styleGAN to manipulate face image attributes and achieve impersonation. Compared to existing methods including gradient-based, patch-based and stealth-based ones, the proposed method aims to result in less visible artifact (to ensure the adversarial perturbation is imperceptible). The proposed method is specific to face recognition and impersonation attack. Experimental evaluations with typical face recognition model and defensively trained face recognition model demonstrate the effectiveness of the proposed method.

**Questions:**

My two major concerns about the paper is written in weaknesses. The attribute editing process is not guaranteed to be orthogonal to identity information. Meanwhile, the proposed method is highly dependent on face generator (styleGAN) and not very flexible to be extended to other domains.

---

The authors have justified the mentioned problems. Some of my original concerns persist, but they largely stem from the underlying GAN.
After rebuttal, I raised the score from 4 to 5.

**Limitations:**

Limitations not clearly elaborated.

**Strengths And Weaknesses:**

# Strengths

1. The proposed method is clearly-motivated, intuitive and effective as demonstrated by experimental results.

# Weaknesses

1. [important; problem setting and evaluation] Is identity information really disentangled from the attribute dimension? Is editing facial attribute really leaving identity information intact? Compared to existing methods, a fair setting should be generating person image with identity information intact while attributes can be changed. If identity information is changed to any extent, it would be much more easier to impersonate. In this regard only several qualitative example is not enough, because the given examples (figure 1 and figure 6) give me a feeling that the resulting person image is not the original identity... especially the third row in figure 6. If the attribute editing is not purely editing attributes, then the comparison in table 1 and table 2 in fact involves unfair comparison, because the other methods like gradient methods do not manipulate identity.

2. [limitation] The proposed method is deeply dependant on the underlying face generation model (style GAN) in terms of both imperceptibility of modification artifacts (quality of generated image), as well as attribute manipulation. Meanwhile, when a high-quality generator is not available in a different domain (like natural images form imagenet), this method will be invalid in that domain.

---

> ### Author Response · Authors · 2022-08-01
> **Response to Reviewer aXGo**
>
> Thanks for your valuable comments and we address your concerns as follows.
>
> **[Q1: Problem setting and evaluation.]**  According to your suggestion, we choose 100 face images from FFHQ and CelebA-HQ and randomly select from these five attributes with different magnitudes to edit the faces by the original StyleGAN [19] for 10 times, and calculate the recognition accuracy between original faces and edited faces by StyleGAN. We find that the recognition accuracy is 100% for all three FR models (i.e., IR152, MobileFace and FaceNet) on both datasets, which indicates that editing facial attributes with StyleGAN [19] does not change its original identity. On the other hand, the aim of our attack is to make FR models recognize the adversarial faces as the targeted person. First, we compute the average cosine similarity between targeted faces and original faces/edited faces by original StyleGAN on FFHQ and CelebA-HQ, as shown in Table 1. It indicates that the original attribute editing hardly affects the similarity to the target face. Furthermore, we calculate the average cosine similarity between targeted faces and edited faces by our attack on FFHQ and CelebA-HQ, as shown in Table 2. The adversarial faces by our attack successfully impersonate the targeted identity with higher cosine similarity.
>
> Table 1: Cosine similarly between targeted faces and original faces/edited faces by original StyleGAN.
>
> | FR model | IR152 | MobileFace | FaceNet|
> |  :----:  | :----:  | :----:  | :----:  |
> | FFHQ          | 0.045/0.043 | 0.168/0.147 | 0.083/0.061 |
> | CelebA-HQ | 0.052/0.049 | 0.179/0.157 | 0.099/0.072 |
>
>
> Table 2: Cosine similarly between targeted faces and edited faces by our attack.
>
> | FR model | IR152 | MobileFace | FaceNet|
> |  :----:  | :----:  | :----:  | :----:  |
> | FFHQ          | 0.231 | 0.306 | 0.412 |
> | CelebA-HQ | 0.237 | 0.306 | 0.429 |
>
> Additionally, Figure C in the supplementary material illustrates a qualitative comparison between edited faces by original StyleGAN and edited faces by our Adv-Attribute attack. As for the third line in Figure 6, the cosine similarity between the original face and the edited face by the original StyleGAN (e.g., 0.675 for MobileNet, and 0.856 for FaceNet) is much higher than the predefined thresholds, which is still recognized as the same identity for three FR models. This demonstrates that the identity information is not changed after editing attributes. Meanwhile, we consider that the selected attributes (e.g., smiling, mustache, blurry and pale skin) may affect the visual quality to some extent. We will add the results and more examples in the revision.
>
> **[Q2(1): Limitation.]** Thanks for your suggestion. We agree with you that the quality of edited faces depends on the synthetic ability of face generators (i.e., StyleGAN [19]) and we will further discuss this limitation in the revised manuscript. Based on the generator, the main contribution of this paper is to integrate the adversary into the semantic information (i.e., face attributes), improving the attack transferability across FR models with more inconspicuous adversarial faces.
>
> **[Q2(1): Other domains.]** While the proposed Adv-Attribute attack is specially designed for face recognition task, the idea of integrating the adversary into the image generation process can be transferred into other domains (e.g., ImageNet classification). Although in some scenarios, the high-quality generator is not available, we can also hide the adversarial noise in the simple image transformation process, including changes in brightness, contrast, etc.

---

> > ### Comment · Reviewer_aXGo · 2022-08-06
> > **Response to Authors**
> >
> > Thanks for the detailed response from authors.
> >
> > Overall, the proposed method manipulates image in the semantics (attribute) space, which can be seen as a new type of perturbation bound compared to the traditional L-infty and L-2. I think even it is hard to conduct the attack in physical setting, demonstrating this conceptually is ok.
> >
> > However, the proposed method heavily depends on a high quality pre-trained generative model, which makes the proposed method inapplicable in domains where there is no such generative model.
> >
> > Secondly, the proposed method also inherits the long-existing issue on evaluating generative models. Numerical evaluation on the generation quality of GAN is always a problem which is not fully aligned with human perception. To my surprise, another reviewer also found that the third row in Figure 6 has switched identity. Human evaluation is also important regardless of the numerical scores. That said, I do not attempt to make the author struggle with a single badly picked example, and I understand that the generated faces are empirically (in most cases) not touching identity information given the provided additional evaluation. Unlike traditional L-p bounds, where we can strictly guarantee that the perturbation will not exceed the bound. In the current problem setting, there is no strict guarantee that the perturbation will not exceed the attribute subspace. This should be clearly clarified in the experiments section, in order to avoid inducing any unfair comparison for future works due to the underlying problems inherited from GANs.
> >
> > I'm willing to raise the score to 5, but no higher, based on the manuscript quality and its potential impact in adversarial defense for face recognition.

---

> > > ### Author Response · Authors · 2022-08-09
> > > **Response to Reviewer aXGo**
> > >
> > > Many thanks for your reply. We agree that no strict guarantee like traditional $\ell_p$ bounds can be used in the attribute subspace for our attack. We will clarify this issue in our revision as you suggest. Additionally, we calculate MSE scores between the original images and our adversarial faces in Table 3, which may serve as an optional evaluation metric to compare the pixel-level variation for different methods.
> > >
> > > We will further polish this paper based on your comments. Thanks again for devoting your time to the careful review.

---

### Official Review · Reviewer_RXXP · 2022-07-19

**Rating:** 4
**Confidence:** 4
**Soundness:** 2 fair
**Presentation:** 2 fair
**Contribution:** 2 fair

**Summary:**

This paper studied the inconspicuous and transferable adversarial attacks on face recognition models. Different from previous works that consider $L_p$ norm perturbations, this paper introduces the framework of adversarial attributes, which generates noise on the attribute space of face images based on StyleGAN. An importance-aware attribute selection approach is proposed to ensure the stealthiness and attacking performance. The experiments on various face models show the effectiveness.

**Questions:**

1. Explain the utility of the proposed method, especially the implementation in the physical world.
2. Clarify the notations and training details.
3. Discuss the semantic inconsistency between real images and the adversarial images.
4. Compare with the SOTA face attacks.

**Limitations:**

The authors have discussed the limitations and negative societal impact of their work.

**Strengths And Weaknesses:**

Strengths:

+ A new method for face attack is proposed, which crafts adversarial noises on the attribute space of face images. The general idea is reasonable and well realized.
+ The framework of face attribute attack is illustrated clearly with some novel techniques, such as important-aware attribute selection and multi-objective optimization.
+ The experiments on typical and robust face recognition models show the effectiveness of the proposed method.

Weaknesses:

- A significant drawback of the proposed method is that it is a digital-world attack method, which can hardly be implemented in the physical world.
- The notations used in this paper are unclear. The paper uses $v_i$ to denote the vicinity appro vector, but what is the exact formulation of $v_i$. I guess $v_i=z_i + n_i$. But why do you adopt the $L_2$ norm of $v_i$ in the stealthy loss in Eq. (5)? I guess it should be the $L_2$ norm of $n_i$?
- How to train the whole framework? Do the StyleGAN models keep fixed or need finetuning? How do you choose the training data?
- The semantics of adversarial face images seem to be inconsistent to the original ones. For the third example in Figure 6, I can hardly recognize the attacker image and adversarial image as the same person.
- The paper did not compare with a state-of-the-art patch attack method [33], which is also based on StyleGAN.

---

> ### Author Response · Authors · 2022-08-02
> **Response to Reviewer RXXP**
>
> Thanks for your valuable comments and we address your concerns as follows.
>
> **[Q1: Utility of our method.]**  Our attack mainly focuses on digital attack, since it can also bring a potential threat to online face recognition (FR) applications, e.g., uploading adversarial photos to impersonate others. Compared with existing digital attacks [6, 16, 24, 27, 33, 36] on FR models, our attack aims to generate more natural and inconspicuous adversarial faces and achieve stronger attack transferability simultaneously. Although some existing methods like wearing AdvGlass [27] and AdvHat [16] implement physical attacks on face recognition, the adversarial patches are visually noticeable and have weak attack transferability across FR models. We consider that both digital and physical attacks are essential for face recognition security, which will draw attention to improving the robustness of FR models.
>
> **[Q2(1): Notations.]** Consistent with your understanding, the vicinity appro vector is defined as $v_i = z_i + n_i$. We will clarify this notation in the revision. Since we do not train or finetune the StyleGAN models, the real attribute vector $z_i$ is a constant vector. Therefore, applying the $ \ell_2 $-norm to $v_i$ can also restrict the magnitude of $n_i$, which is equivalent to applying the $ \ell_2 $-norm to $n_i$.
>
> **[Q2(2): Training details.]** During the whole process, we remain the StyleGAN model fixed and thus assign no training data to StyleGAN model. As for the training of adversarial noise generators, we first ensure the targeted faces and randomly choose face images with other identities from FFHQ or CelebA-HQ as training data. The attribute noise generators are trained using Adam optimizer with an initial learning rate 0.0001. More experimental setup can be referred to Section 4.1.
>
> **[Q3: Semantic inconsistency.]**  The original StyleGAN itself has slight impacts on the visual quality of edited faces. However, we calculate that the recognition accuracy between original faces and edited faces by the original StyleGAN is 100% for all three FR models on both FFHQ and CelebA-HQ, which indicates that editing attributes with original StyleGAN does not change its original identity. Besides, our attack depends on the original StyleGAN to generate adversarial faces, thus the quality of adversarial faces is limited by the synthetic ability of StyleGAN. In the supplementary material, Figure C shows original source faces, edited faces by the original StyleGAN, and edited faces by our attack (including the third example in Figure 6). And the image quality of edited faces by our attack is close to the ones by the original StyleGAN.
>
> We consider that the multiple selected attributes like smiling, mustache, blurry and pale skin could impact the visual semantic inconsistency to the original face, while the FR models still recognize them as the same identity. Moreover, compared with existing stealthy-based attacks (e.g., Adv-Face [6], Adv-Makeup [26] and Semantic-Adv [24]), our method generates more inconspicuous adversarial faces and further strengthens the attack transferability across FR models.
>
> Thanks for your suggestion, we will add more discussions in the revision.
>
> **[Q4: Compared with GenAP [33].]** Strictly following the settings of GenAP [33], we select eye regions to craft the adversarial patches. The tables below compare the attack transferability between GenAP [33] and our attack on FFHQ and CelebA-HQ, using the same FR models to attack both basic models and robust models. In general, our method performs better transfer attacks on all datasets compared with GenAP. Meanwhile, the edited faces by our attack are more inconspicuous than the adversarial patches of GenAP. We will supply the complete experimental results and visualizations in the revision.
>
> Table 1: ASR results of Gen-AP and our attack against basic models.
>
> FFHQ:
> | Target model | IR152 | MobileFace | FaceNet |
> |  :----:  | :----:  | :----:  | :----:  |
> | Gen-AP | 12.00 | 19.90 |   8.20 |
> | Ours      | 44.30 | 50.20 | 31.80 |
>
>  CelebA-HQ:
> | Target model | IR152 | MobileFace | FaceNet |
> |  :----:  | :----:  | :----:  | :----:  |
> | Gen-AP | 19.50 | 24.40 | 15.80|
> | Ours      | 46.30 | 49.90 | 31.90|
>
> Table 2: ASR results of Gen-AP and our attack against robust models.
>
> FFHQ:
> | Training model | -IR152 | -IR152 | -MobileFace | -MobileFace | -FaceNet | -FaceNet |
> |  :----:  | :----:  | :----:  | :----:  | :----:  | :----:  | :----:  |
> | Target model | AR | TR | AR | TR | AR | TR |
> | Gen-AP | 36.50 | 13.70 | 34.00 | 10.60 | 27.70 |   9.80 |
> | Ours      | 60.40 | 30.10 | 53.30 | 26.30 | 63.00 | 30.10 |
>
>  CelebA-HQ:
> | Training model | -IR152 | -IR152 | -MobileFace | -MobileFace | -FaceNet | -FaceNet |
> |  :----:  | :----:  | :----:  | :----:  | :----:  | :----:  | :----:  |
> | Target model | AR | TR | AR | TR | AR | TR |
> | Gen-AP | 46.00 | 15.80 | 45.30 | 15.70 | 44.90 | 15.30 |
> | Ours      | 60.80 | 33.50 | 56.60 | 34.00 | 61.60 | 34.10 |

---

> > ### Comment · Reviewer_RXXP · 2022-08-07
> > **Thanks for the response**
> >
> > Thanks for the response to my questions. The authors clarified some of my concerns, including:
> >
> > 1. Q1 - utility of the attacks: the authors specify the scope of this paper for digital-attacks, which is ok to me.
> > 2. Q2 - notations and training details: this has been clarified. The authors should further revise the paper to make them clearer.
> > 3. Q4 - compare with GenAP: thanks for the new results. It seems that the proposed method is more effective. Please add the results in the revision.
> >
> > However, I am still not convinced by the response to Q3. The authors have stated that "Besides, our attack depends on the original StyleGAN to generate adversarial faces, thus the quality of adversarial faces is limited by the synthetic ability of StyleGAN." Could you try more effective generative models? The current results in the third example in Figure 6 make me doubt about the performance. Do you cherry-pick the visualization results?

---

> > > ### Author Response · Authors · 2022-08-09
> > > **Response to Reviewer RXXP**
> > >
> > > Many thanks for your reply and we will further address your concerns as follows.
> > >
> > > **[Q3: Semantic inconsistency.]**  In the revised supplementary material, we provide more qualitative results from the FFHQ and CelebA-HQ datasets. Figure E and Figure F compare the original source faces, the edited faces by original StyleGAN [19] and the edited faces by our attack. In general, the majority of adversarial edited faces achieve favourable visual quality and attacking performance, while only several examples (e.g., the third row in Figure F) are slightly semantically-inconsistent with the original image for human observers.
> > >
> > > In order to explore the factors that affect the visual quality in our method, we firstly apply our attack with a more advanced face generator (i.e., HFGI [A]) as you suggested. Since the official implementation of HFGI only provides five off-the-shelf attributes in the edited spaces (i.e., smile, age, lip, bread and eyes), which are not consistent with the setting of our attack, we choose bread and smile as the selected attributes during attacking for the fair comparisons. Figure G in the revised supplementary material compares the visual quality of adversarial faces by HFGI [A] and StyleGAN [19]. The numbers below the images are the cosine similarity scores between the targeted faces and crafted faces. We observe that HFGI [A] slightly improves the image quality and achieves comparable attacking performance with our original generator [19]. It indicates that our attack method does not rely on the specific model and could be deployed to various generative models. Due to the limited time, we will add the complete experimental results by HFGI [A] in the revision.
> > >
> > > On the other hand, we further explore the semantic inconsistency through analyzing the attribute editing space for the third example in Figure 6. We observe the visual effects on adversarial faces with different selected attributes and find that the pale face is not fully disentangled in StyleGAN [18], leading to the semantic inconsistency in the third example of Figure 6. Thus, we try to remove this attribute and utilize the rest attributes as the editing spaces to generate the adversarial face by our attack. Figure D compares the adversarial faces under original settings and the adversarial faces without pale face. The crafted adversarial example without pale face is much closer to the original face for human observers. We will supply these discussions and analyses in the revised paper.
> > >
> > > Thanks again for devoting your time to the careful review.
> > >
> > > Reference:
> > >
> > > [A] High-Fidelity GAN Inversion for Image Attribute Editing, In CVPR, 2022.

---

### Meta-Review · Area_Chair_qWru · 2022-08-27

**Recommendation:** Accept
**Confidence:** Certain

**Metareview:**

This paper studies adversarial attacks on facial recognition systems. The key contribution is that, instead of directly manipulating pixel space, this paper proposed to perturb the facial attributes for generating inconspicuous and transferable adversarial examples. The initial concerns are mostly about requiring 1) more ablations/comparisons, and 2) clarifications on experiment details and visualization (especially Figure 6).

Most concerns are well addressed in the rebuttal, and 3 (out of 4) reviewers agree to accept this paper. The reviewer RXXP is still (slightly) concerned about the novelty contribution and rates it as a borderline case. Given its effectiveness and comprehensive analysis, the AC agrees that this paper has its own merits and will be of interest to the general NeurIPS community, therefore recommend accepting it.

In the final version, the authors should include all the clarifications and the additional empirical results provided in the rebuttal.



**Award:**

No

---

### Decision · Program_Chairs · 2022-09-14

Accept